# The Role of Lithium in the Aging Precipitation Process of Al-Zn-Mg-Cu Alloys and Its Effect on the Properties

**DOI:** 10.3390/ma16134750

**Published:** 2023-06-30

**Authors:** Jing-Ran Sun, Bai-Xin Dong, Hong-Yu Yang, Shi-Li Shu, Feng Qiu, Qi-Chuan Jiang, Lai-Chang Zhang

**Affiliations:** 1Key Laboratory of Automobile Materials, Ministry of Education and Department of Materials Science and Engineering, Jilin University, Renmin Street NO. 5988, Changchun 130025, China; sjr22@mails.jlu.edu.cn (J.-R.S.); jiangqc@jlu.edu.cn (Q.-C.J.); dongbx20@mails.jlu.edu.cn (B.-X.D.); 2School of Materials Science and Engineering, Jiangsu University of Science and Technology, Zhenjiang 212003, China; 3School of Mechanical and Aerospace Engineering, Jilin University, Renmin Street NO. 5988, Changchun 130025, China; shushili@jlu.edu.cn; 4Chongqing Research Institute, Jilin University, Chongqing 401123, China; 5Centre for Advanced Materials and Manufacturing, School of Engineering, Edith Cowan University, 270 Joondalup Drive, Joondalup, Perth, WA 6027, Australia

**Keywords:** Al-Zn-Mg-Cu-Li alloy, aging behavior, precipitated phase, Li content, service properties

## Abstract

It is well known that the development of lightweight alloys with improved comprehensive performance and application value are the future development directions for the ultra-high-strength 7xxx series Al-Zn-Mg-Cu alloys used in the aircraft field. As the lightest metal element in nature, lithium (Li) has outstanding advantages in reducing the density and increasing the elastic modulus in aluminum alloys, so Al-Zn-Mg-Cu alloys containing Li have gained widespread attention. Furthermore, since the Al-Zn-Mg-Cu alloy is usually strengthened by aging treatment, it is crucial to understand how Li addition affects its aging precipitation process. As such, in this article, the effects and mechanism of Li on the aging precipitation behavior and the impact of Li content on the aging precipitation phase of Al-Zn-Mg-Cu alloys are briefly reviewed, and the influence of Li on the service properties, including mechanical properties, wear resistance, and fatigue resistance, of Al-Zn-Mg-Cu alloys are explained. In addition, the corresponding development prospects and challenges of the Al-Zn-Mg-Cu-Li alloy are also proposed. This review is helpful to further understand the role of Li in Al-Zn-Mg-Cu alloys and provides a reference for the development of high-strength aluminum alloys containing Li with good comprehensive properties.

## 1. Introduction

The soaring requirements of aerospace industries have greatly propelled the advancement of novel materials with high mechanical properties, light weight, and high strength. Developed in the 1920s and 1930s, the ultra-high-strength 7xxx series aluminum alloys, namely, Al-Zn-Mg-Cu series aluminum alloys, exhibit high strength, favorable resistance to fatigue and corrosion, and a low crack growth rate after aging strengthening [1,2,3]. The 7xxx series aluminum alloys have become high-quality structural materials used in aviation (Figure 1) and transportation fields, and they are the ideal choice for the development of high-performance lightweight metal materials [1,4,5]. Although 7xxx series aluminum alloys have many advantages, their high density and poor plasticity limit their applications to some extent [6]. Further lightening, improving the specific strength and specific stiffness, increasing the elastic modulus, and improving the comprehensive performance and application value of Al-Zn-Mg-Cu alloys are the focus of developing high-performance aluminum alloys. Among the focused methods, reducing the density is an important approach.

Al-Li alloy possesses several advantages, including low density, high elastic modulus, high specific strength, and excellent corrosion resistance [7,8,9]. In the aerospace industry, using Al-Li alloy instead of conventional aluminum alloys can reduce weight, save fuel, and save costs. The research of adding a certain amount of Li to commonly used aluminum alloys to exploit the benefits of Al-Li alloys has received extensive attention. In the past decades, Li-containing aluminum alloys such as Al-Cu-Li [10,11,12], Al-Cu-Mg-Li [13], and Al-Mg-Li [14,15] have been studied, and Al-Zn-Mg-Cu-Li alloys have also garnered significant interest in recent years [16].

Li (with a density of 0.54 g/cm^3^) is the lightest metal in nature. Adding Li to aluminum alloys can reduce the alloy density, increase the elastic modulus, and significantly decrease the weight of the base alloy while improving performance [17,18]. According to Ref. [19], the addition of 1 wt.% of Li to a pure aluminum alloy results in a reduction in density of 3% and an increase in elastic modulus of 6%. Moreover, at intermediate levels of stress intensity, element Li improves the resistance of the alloy to fatigue crack growth [20]. Another benefit is that aluminum alloys incorporating Li respond to age hardening [21]. In addition, Li has an excellent solid solubility (up to 4 wt.% at 610 °C) in aluminum, which can have a good solid solution strengthening effect. Wang et al. [22] added Li to 7050 aluminum alloy by melt casting and found that the addition of Li can reduce the alloy density, improve the mechanical properties of the alloy, e.g., increased elastic modulus and specific strength, improve the fracture mode, and reduce the crack extension rate, as well as enhance the corrosion resistance. Adding Li in a certain range can also refine grains, improve grain morphology, and significantly enhance the hardness of the alloy. Bai’s team [23] pointed out that the properties of 7xxx aluminum alloys are strongly associated with the aging precipitation process after adding appropriate amounts of Li. The incorporation of suitable quantities of Li elements into the alloy can change the aging precipitation process and the microstructure of the precipitated phases, the size, shape, distribution and volume fraction of the original precipitated phases in the aluminum alloy and generate new reinforced phases, thereby significantly improving the microstructure and properties of the alloy. As such, adding an appropriate amount of Li to the aluminum alloy can not only reduce the density, improve the elastic modulus, and allow a lighter weight by taking advantage of the aluminum–lithium alloy, but, also importantly, the addition of a Li element affects the aging precipitation process of the aluminum alloy and changes the precipitating phase to improve the properties of the alloy [24,25,26]. Especially for the ultra-high-strength 7xxx series Al-Zn-Mg-Cu alloys that can be aged, it is of significance to study the effect of the addition of Li on the properties of the alloy.

In this article, the effect of Li on the aging precipitation behavior of Al-Zn-Mg-Cu alloys and the effect of Li content on the precipitation phase of the alloy and its mechanism are reviewed. The effects of Li on the Al-Zn-Mg-Cu alloy and its mechanical properties, wear resistance, and fatigue resistance are expatiated. It helps the reader to systematically understand the role of Li in the aging precipitation process of Al-Zn-Mg-Cu alloys, so that the advantages of Li can be maximized by regulating the Li content and the precipitation process of Li-containing alloys to obtain Al-Zn-Mg-Cu alloys with excellent overall performance. Finally, the current problems are highlighted and the prospects for development and some potential research orientations in the future are also put forward. It is of reference significance for the development of high-performance Al-Zn-Mg-Cu alloys.

## 2. Effect of Li on the Aging Behavior and Mechanism of Al-Zn-Mg-Cu Alloy

Al-Zn-Mg-Cu alloys achieve high performance through complex heat treatment processes such as solution treatment, quenching, and multi-stage aging [25], while Li has a sufficiently high vacancy binding energy (0.25–0.26 eV) to preferentially capture excess vacancy and impede vacancy condensation and annihilation during the process of quenching and aging [27]. Therefore, the addition of Li to the Al-Zn-Mg-Cu alloy can not only change the Guinier–Preston zone (GP zone) aggregation kinetics, aging precipitation behavior, and the hardening response of the alloy, but also significantly impact the type and morphology of the precipitation phase of the aluminum alloy, the aging precipitation process, and the microstructure of the alloy, and thus influence the properties of the alloy [16,28].

### 2.1. Effect of Li on the Aging Precipitation GP Zone

7xxx series aluminum alloys are precipitation-hardening Al-Zn-Mg-Cu alloys, and the typical precipitation sequence during the aging treatment can be briefly summarized as: supersaturated solid solution→GP zones→η′ phase→η phase [29] (η is formed by the transformation of η′, and η′ is obtained by the transformation of the GP zone). Research has shown that two types of GP zones are created during the low-temperature aging process of 7xxx series alloys [30,31,32], i.e., spherical GP(Ⅰ) formed by Mg and Zn aggregation at room temperature, and disk-like GP(Ⅱ) synthesized by solute atoms and vacancy unity above 343 K (Figure 2a–f). Among them, the formation of the GP(I) zone is controlled by thermodynamic conditions and is almost independent upon vacancy concentration in the matrix; by contrast, GP(Ⅱ) is directly related to vacancy concentration in aluminum alloys. Because of the significant influence of Li atoms on vacancy distribution [33], the precipitation pattern of the precipitated phase and the formation of the GP zone are closely related to adding content to the alloys [26].

Wei et al. [34] quantitatively examined the influence of Li on the GP zone of Al-Zn-Mg-Cu alloy using Differential Scanning Calorimetry (DSC) experiments, and they concluded that the impact of Li on the precipitation of the microstructure of the alloy is associated with its content. The addition of Li in the alloy can inhibit the conversion of the GP(Ⅱ) zone and reduce its content, while increasing the content of the GP(I) zone. When the content of Li in the alloy is low (generally lower than 1.2 wt.%), the precipitation pattern of the phase is consistent with that of an Al-Zn-Mg-Cu alloy. In such a situation, the effect of Li on 7xxx series alloys is mainly to reduce the solid solubility of Zn and Mg in the matrix. Since Li atoms solidly dissolved in the matrix has higher vacancy binding energy than Zn or Mg atoms, the Li vacancy cluster is preferentially formed during solution treatment and quenching, which reduces the concentration of free vacancy, minimizes the chance for Mg and Zn to encounter vacancy, promotes the aggregation of Zn and Mg atoms, and thus promotes the generation of the GP(I) zone. At the same time, the reduction in the free vacancy concentration hinders the diffusion of Zn and Mg atoms, thereby inhibiting the formation of the GP(Ⅱ) zone and the coarsening of the η′ phase [35]. Due to the small amount of Zn and Mg in the matrix, Zn and Mg atoms are prone to aggregate so that the dissolution activation of the GP region is slightly lower [34]. Wei et al. [36] studied the aging precipitation behavior of 7075 aluminum alloy and the impact of Li on the microstructure transformation during the aging of this alloy using differential thermal analysis, and also calculated the dissolution kinetic parameters of the GP zone and η′ phase in 7075 aluminum alloy with and without Li, respectively. Their findings indicated that Li had no significant effect on the dissolution activation energy of the GP zone, but could significantly increase the dissolution activation energy of the η′ phase (Figure 2g,h). This is mainly attributed to the dissolution of the η′ phase requiring the diffusion of solute atoms, which is closely related to the free vacancy concentration. In 7075 alloy containing 1.0 wt.% Li, Li can bind a significant quantity of vacancies and inhibit the spread of solute atoms, so that the dissolution temperature of the η′ phase lags behind that of the 7075 alloy, and the dissolution activation energy of the η′ phase is significantly higher than that of the 7075 alloy. Therefore, the addition of Li mainly takes advantage of its high binding energy with vacancies to influence the vacancy distribution and vacancy concentration through the combination of Li with vacancy, and then affects the aging precipitation of the GP zone, thereby promoting the formation of the GP(Ⅰ) zone and inhibiting the formation of the GP(Ⅱ) zone, as shown in Figure 3.

**Figure 2 materials-16-04750-f002:**
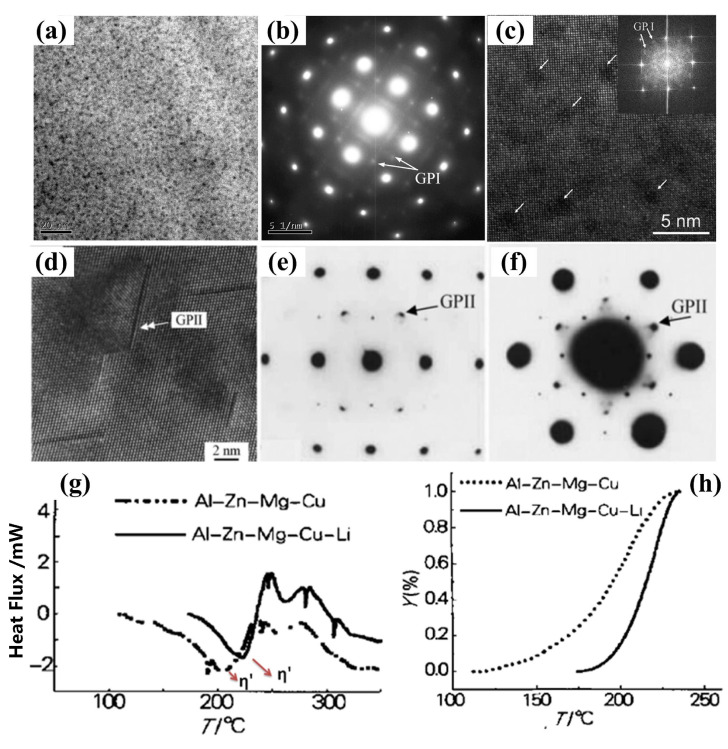
(**a**) and (**b**) TEM images and corresponding selected area electron diffraction (SAED) patterns of GP(Ⅰ) zones of the 7050 Al alloy [37]; (**c**) high-resolution TEM (HRTEM) [38]; (**d**) HRTEM images of GP(II) zones; (**e**) and (**f**) SAED patterns along different crystal surfaces in Al-Zn-Mg alloys [31]; (**g**) DSC results of double alloys; (**h**) (dY/dT)v s temperature curves for η′ phase dissolution of two alloys [36].

**Figure 3 materials-16-04750-f003:**
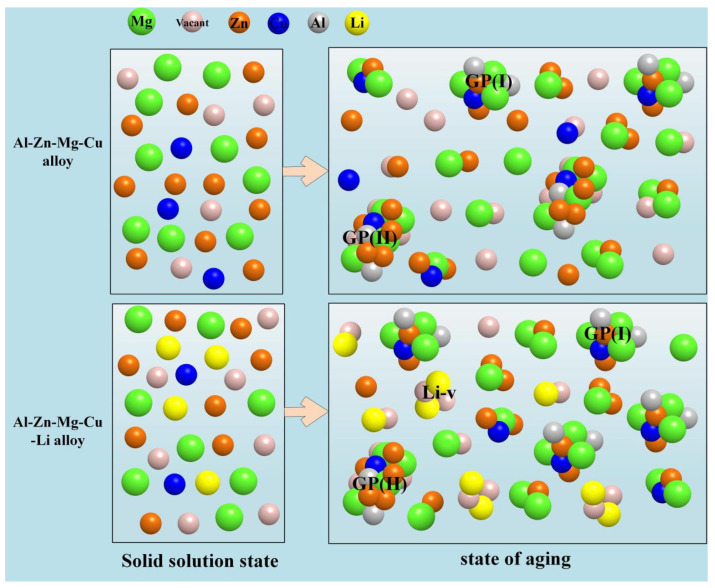
Effect mechanism of lithium on GP zone precipitation of Al-Zn-Mg-Cu alloy during aging precipitation [34,35,36].

### 2.2. Effect of Li on the Aging Precipitation Phase

The mechanical performance of age-strengthened aluminum alloy largely depends on the precipitates formed by aging, which is the basic strengthening mechanism of aluminum alloys [39]. In addition to reducing the density of aluminum alloys, refining grains, and playing a solid solution strengthening role in aluminum alloys, Li in Li-containing aluminum alloys mainly influences the alloy properties as a component element of the strengthening phase precipitated by aging [27]. It is shown that the addition of Li in Al-Zn-Mg-Cu alloys affects the distribution of vacancies, changes the nucleation of precipitated phases, affects the size, shape, distribution, and volume fraction of the original precipitated phases, and changes the width of precipitation-free zones (PFZs). Furthermore, a certain amount of Li can be used as a strengthening phase constituent element to generate new strengthening phases, which affects the strength and plastic toughness of the alloy.

#### 2.2.1. Common Precipitated Phase in Li-Containing Aluminum Alloys

During the age-hardening process, the aging-precipitated phases in the Al-Cu and Al-Li binary alloys and in the Al-Cu-Li and Al-Mg-Li ternary alloys may appear in the Al-Zn-Mg-Cu-Li alloy, resulting in the complex phase composition of the alloy. Table 1 displays the typical aging-precipitated phases in Al-Zn-Mg-Cu-Li alloys during aging treatment. The structure images of these phases are presented in Figure 4.

**Table 1 materials-16-04750-t001:** Common precipitated phases in Al-Zn-Mg-Cu-Li alloy.

Precipitation Phase	Stoichiometric Ratio	Crystal Structure	Lattice Parameters	Common Shapes	Ref
η′ phase	MgZn_2_	Hexagonal crystal system	a = 0.496 nm	Disc-shaped	[30]
η phase	MgZn_2_	Hexagonal crystal system	a = 0.522 nm	Multiple shapes	[40]
δ′ phase	A1_3_Li	Ordered fcc	a = 0.401 nm	Spherical or lenticular	[41]
T phase	(Al,Zn)_49_Mg_32_	body-centered cube	a = 1.416 nm	Icosahedral sphere	[42]
S phase	Al_2_CuMg	Cmcm	a = 0.400 nm	lath-shaped	[43]

**Figure 4 materials-16-04750-f004:**
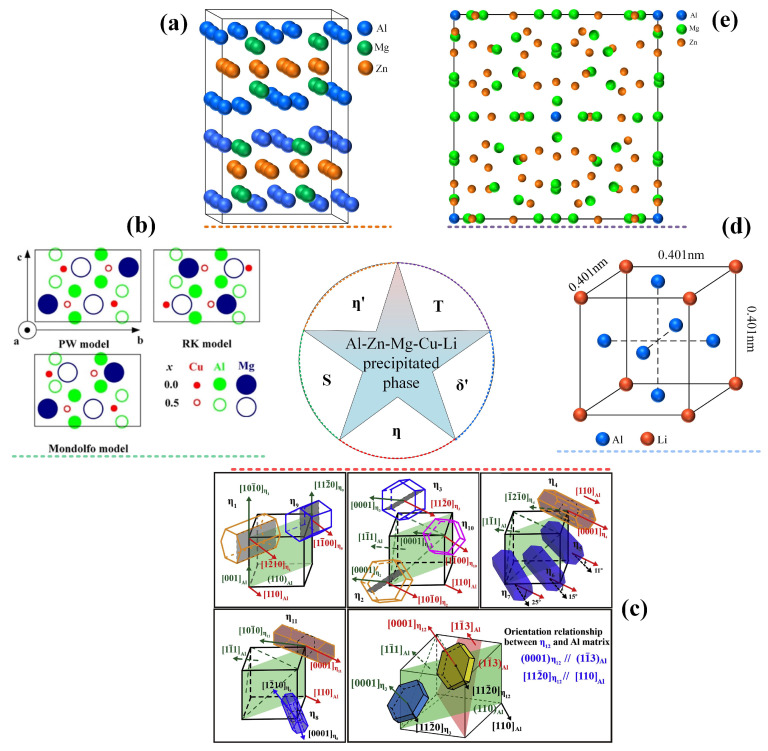
(**a**) η′ phase structure diagram; (**b**) S phase three-structure model [43]; (**c**) mechanism of η phase [44]; (**d**) structure of δ′ phase; (**e**) T phase structure diagram [45].

1.η′ phase (MgZn_2_)

The η′ phase is the primary reinforcing phase in Al-Zn-Mg-Cu alloy and it plays a major role in determining the strength of the alloy [30]. It is commonly believed that the η′ phase evolves from the GP zone during aging, and the η′ phase is disk-shaped with a thickness of 7 or 11 atomic layers, which are projected in different directions, as illustrated in Figure 5a. The crystal structure of the η′ phase is still controversial because the η′ phase, as an intermediate transition phase in the Al-Zn-Mg-Cu alloy, usually has a transformation process. At present, a more consistent view is that the η′ phase has a hexagonal structure with a space group of P6_3_/mmc (a = 0.496 nm, c = 1.405 nm) [40,46,47,48], and the η′ phase is fully coherent with the α-Al matrix, which is mainly manifested by the co-lattice of the broad side of the disk with the {111}_Al_ side of the Al matrix rather the side of the disk, as illustrated in Figure 5b [40]. As the process of aging persists, the metastable η′ phase is transformed into a stable η phase [49].

**Figure 5 materials-16-04750-f005:**
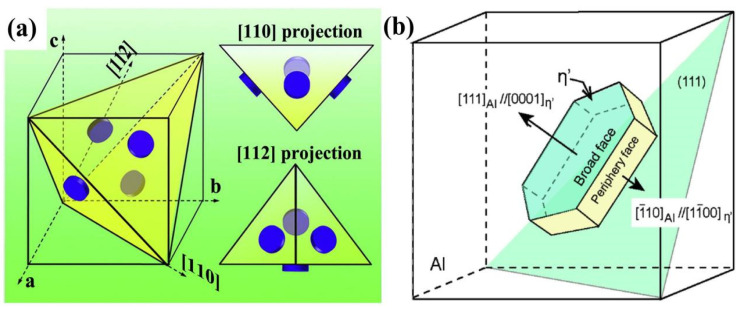
(**a**) Three-dimensional schematic diagram of disk-precipitated phase on {111}_Al_ surface under different projections [50]; (**b**) schematic diagram of the relationship between disk-like η′ phase and Al matrix orientation [40].

2.η phase (MgZn_2_)

The η phase is an equilibrium stable phase, mainly in the form of rods or slats, which is completely incoherent with the matrix and contributes very little to the strength of the alloy [21]. According to literature, the η phase has a hexagonal structure with a space group of P6_3_/mmc (a = 0.504 nm, c = 0.828 nm) and an atomic ratio of Zn/Mg higher than 1.5 [44,51,52]. The η phase is usually transformed from the η′ phase and can also be precipitated directly from the matrix, which has a variety of morphologies and a multitude of orientation relationships with the matrix (Figure 4c) [43]. Generally, the η phase is MgZn_2_, but it also usually contains a certain elemental content of Al due to the energy of the Al atoms partially replacing the Zn and Mg atoms. The η phase has lost its coherent relationship with the matrix, and the loss of the strengthening interface softens the aged alloy compared to the GP zone and a series of metastable phases in the early stages of aging [21].

3.δ′ phase (Al_3_Li)

The δ′ phase is a metastable phase with Al_3_Li composition, and the Al_3_Li is an ordered LI_2_-type (Cu_3_Au-type) superlattice structure with a lattice constant of a = 0.401 nm and a space group of Pm3m [53,54,55]. As an ordered face-centered cubic (fcc) structure, Li atoms occupy the vertex position of the fcc structure, and Al atoms occupy the face-center position. The orientation relationship between Li and matrix is (100)Al_3_Li//(100)Al, [100]Al_3_Li//[100]Al [41]. The δ′ phase is completely coherent with the Al matrix [54], and the mismatch with the Al matrix is very small even if the δ′ phase grows larger. The δ′ phase is usually spherical and has a small interfacial energy [53]. This low lattice distortion and the interfacial energy with the matrix make the δ′ phase tend to precipitate. However, the solute Li in δ′ is easy to combine with the vacancy, and the solubility of Li is sensitive to temperature changes and has high mobility, making the δ′ phase unstable [56]. Studies showed that the δ′ in a spherical shape is prone to be cut in pairs by dislocations during the plastic deformation of the alloy, leading to co-planar slip and causing stress concentration at grain boundaries. This induces crack sprouting at grain boundaries and rapid expansion along the grain boundaries or slip surface [54,57], thereby decreasing the ductility and fracture toughness of the alloy and brittle fracture occurs [41,54]. In addition, as the aging time is extended, the coarsening of the δ′ phase decreases the concentration of Li atoms in the vicinity of the high-angle grain boundary, resulting in the appearance of the precipitation free zone (PFZ), and the precipitation of the δ phase (equilibrium phase of the δ′ phase) at the grain boundary, which has a detrimental impact on the strength and toughness of the alloy [58]. Therefore, the coarse δ′ phase should be avoided during the composition design and heat treatment process formulation of Li-containing aluminum alloys.

4.T phase ((Al,Zn)_49_Mg_32_)

T-phase ((Al,Zn)_49_Mg_32_) is recognized as possessing a body-centered cubic structure with the space group Im3 (No. 204) and with the lattice parameter a = 1.416 nm [42,45,59]. The unit cell of this phase comprises 162 atoms (Figure 4e) [59]. The T′ phase is the precursor of the T phase with a morphology resembling spherical atomic clusters. The TEM image of the T′ phase and the corresponding diffraction pattern are exhibited in Figure 6d–f [21]. The atomic structure of the T-phase is composed of a characteristic icosahedral sphere with no atoms in the center [59], and Zn and Mg atoms occupy random positions. Researchers believe that the dissimilarities in the structure and composition between the T′ and T phases are so small that they are difficult to distinguish, and thus no longer make accurate distinctions. Usually, the T-phase is not the phase obtained by conventional heat treatment, and most researchers consider the T-phase to be a high-temperature phase. The lattice consistent of T is c = 0.87~0.90, which is marginally larger than that of η′ with c = 0.868, and their pattern and distribution exhibit a high degree of similarity [60]. Gayle et al. [61] pointed out that it is hard to differentiate between T and η′, except when using a high-resolution electron microscope (HREM).

5.S phase (Al_2_CuMg)

The S phase is composed of Al_2_CuMg and forms as laths along (100)_Al_, with {012}_Al_ habit. Figure 6h presents a representative HREM picture of the S phase along the [100]_S_ and [010]_S_ directions [62]. However, measurements of the crystal structure of the S phase are still controversial. Concerning the crystal structure of Al_2_CuMg, several models have been proposed. The initial S phase model was presented by Perlitz and Westgren (PW) [63] based on X-ray diffraction. The unit cell in the PW model is orthorhombic with dimensions a = 0.400 nm, b = 0:923 nm, and c = 0.714 nm, space group Cmcm, containing 16 atoms in the ratio Al:Cu:Mg = 2:1:1. Subsequently, the Mondolfo model and RK model were introduced by Mondolfo [64] and Radmilovic et al. [62,65], as depicted in Figure 4b. However, the PW model is currently the most recognized model [43]. The S phase (Al_2_CuMg) and S′ exhibit a near-identical composition, except for a slight variance in their respective lattice parameters. Therefore, many authors do not make any distinction between these two orthorhombic phases. However, the S′ phase provides significant strengthening in Al-Cu-Mg-based alloys, which exhibit poor precipitate ratios in Al-Li systems [41]. Duan et al. [25] proposed that the inclusion of Li may have an impact on the process of nucleation and growth of S′, leading to a reduction in the precipitation of S′ in alloys. TEM images of the η′ phase, T′ phase, S phase, and δ′ phase are shown in Figure 6, respectively.

**Figure 6 materials-16-04750-f006:**
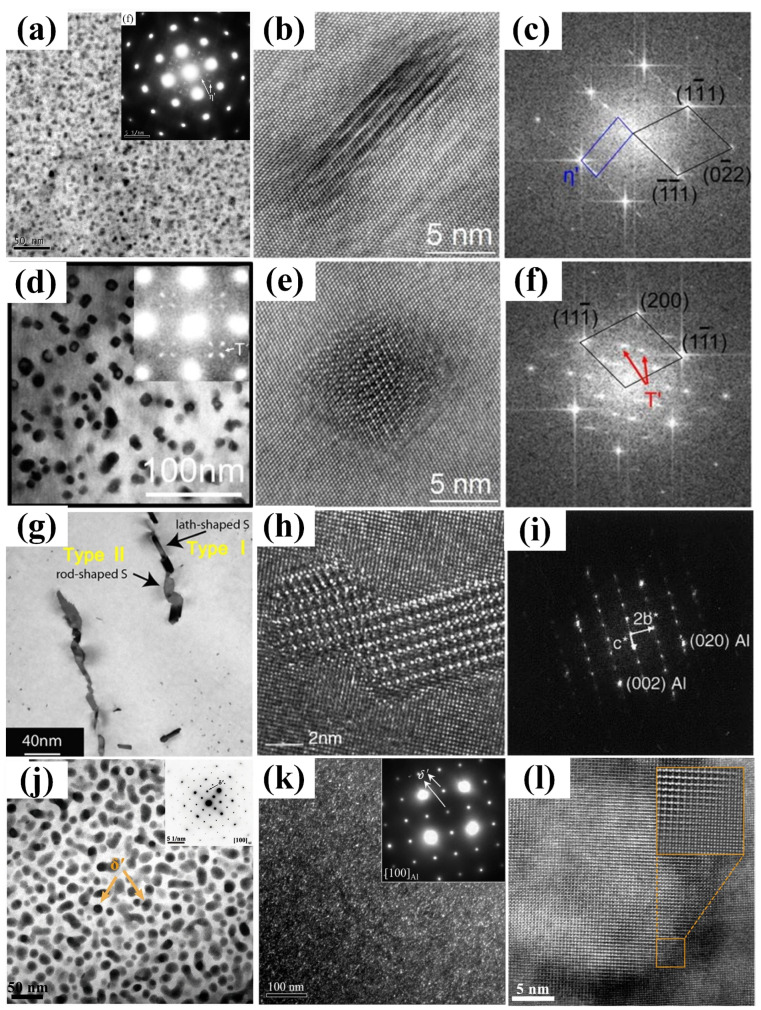
(**a**) TEM images of η′ [37]; (**b**) and (**c**) HRTEM and SAED patterns of η′ [66]; (**d**) TEM images of T′ [45]; (**e**,**f**) HRTEM and SAED patterns of T′; (**g**) TEM images of S [67]; (**h**) HRTEM images and (**i**) SAED patterns of S [68]; (**j**–**l**) TEM images, corresponding SAED patterns, and HRTEM images of δ′. Schematic diagram of the relationship between disk-like η′ phase and Al matrix orientation [55].

#### 2.2.2. Effect of Li Content on the Precipitated Phase

The addition of Li elements to the alloy affects the size and distribution of the aging precipitation phase, and the Li content is key to the type of precipitation phase. Bai et al. [23] concluded that Li-containing alloys have finer precipitation-reinforced phases and narrower precipitation-free zones (PFZs) than Li-free alloys by comparing the double-stage aging treatment of two 7xxx series alloys with and without Li. Gregson et al. [69] demonstrated that the Li content in the alloy is an important factor that affects the precipitation phase and age-hardening behavior. In Al-Zn-Mg-Cu alloys, Mg vacancy aggregation is dominant, along with Zn vacancy, Cu vacancy, and the composite aggregation of Zn-Mg, and η (MgZn_2_), θ (Al_2_Cu), S (Al_2_CuMg) and other precipitation phases are obtained after aging treatment, but the main aging precipitation products are η phases due to the high content of Zn and Mg elements in the alloy. The addition of Li elements to the alloy has a great influence on the vacancy distribution, and numerous studies have demonstrated that the addition of Li reduces the concentration of free vacancies (namely, the concentration of vacancies) in the alloy and hinders the diffusion of Zn and Mg atoms. When increasing the Li content, fewer free vacancies combined with other atoms decrease until the η′ (MgZn_2_) phase cannot be precipitated [33]. Xie et al. [70] showed that with the increase in the Li content (0 wt.%, 0.5 wt.%, 1 wt.%, and 2 wt.%) in 7075-Li alloy, the diffraction peaks of the η (MgZn_2_) phase decrease continuously, and the diffraction peaks of the δ′ and δ phase increase continuously, while the η phase disappears when the Li content is 2 wt.%, as shown in Figure 7a–c. The increase in the Li content contributes to the formation of δ′ and δ phases while inhibiting the formation of the η phase. Huang et al. [27] showed that the addition of 0.7 wt.% Li to 7075 alloy results in vacancy-rich GP zones, T′ and T [(Al,Zn)_49_Mg_32_] phases, which are different from those of 7075 alloy; by contrast, when adding 2.0 wt.% Li to 7075 alloy, aging precipitation is dominated by the δ′ (Al_3_Li) phase, and the coarse T [(Al,Zn)_49_Mg_32_] phase replaces the original fine η′ (MgZn_2_) phase. Zhao et al. [71] showed that the η′ phase or T′ phase precipitates on the matrix of Al-Zn-Mg-Cu alloys with no more than 1.0 wt.% Li, and high hardness can be achieved with secondary or multi-stage aging. However, the main precipitated phases in the Al-Zn-Mg-Cu alloys with higher than 1.8 wt.% Li after aging treatment are the δ′ (Al_3_Li) phase, S (Al_2_CuMg) phase, and T phase, which results in poor plasticity and toughness. In view of the detrimental effect of the precipitation phase on alloy properties during the aging of aluminum alloys with high Li content, most researchers have focused on alloys with 0.9~1.3 wt.% Li content to further improve the alloy properties.

When low-content (generally below 1.2 wt.%) Li is added to the Al-Zn-Mg-Cu alloy, Li, as a trace element, mainly exists in the GP zone or in solid solution, which only affects the morphology of its precipitated phases and does not change the order and type of precipitated phases; therefore, the strengthened phases in the alloy are still dominated by η′ without forming coarse δ′ (Al_3_Li) phases. At the same time, the high vacancy binding energy with Li atoms allows Li atoms to bind to large number of vacancies and, therefore, Li vacancy pairs form preferentially. This inhibits the uniform nucleation of the Zn-rich phase and slows down the diffusion of Zn and Mg atoms, hindering the growth of the first precipitated η′ (MgZn_2_) phase, which, in turn, refines the η′ phase and makes the η′ phase diffusely distributed. Moreover, due to the reduction in free vacancies, the chance of diffusion to the grain boundary is also smaller, making the precipitation-free zone at the grain boundary narrower. Zheng et al. [72] studied the precipitated phases in an Al-Zn-Mg-Cu-1.0 wt.% Li alloy under various aging conditions by small-angle X-ray scattering and TEM and concluded that the predominant precipitates in this alloy are η′ rather than δ′ based on the selected area electron diffraction (SAED) shown in Figure 7d–g. Zhou et al. [20] investigated the microstructure and properties of Al-Zn-Mg-Cu alloy with 1.01 wt.% Li and confirmed the precipitation of a fine and dispersed η′ (MgZn_2_) phase in the alloy using XRD (Figure 7j). The strength of η′(MgZn_2_) is weak, so the particles are very small. Sodergren et al. [73] analyzed the precipitation phase in the alloy using DSC and TEM. Their results showed that the AA7029 alloy containing 0.4 wt.% Li exhibited the same microstructure as the AA7029 alloy, i.e., the GP zone and η′ mixture (heated to the first peak of the DSC curve) and the equilibrium η on the grain boundary (MgZn_2_) and the mixture of η′ and η in the matrix when heated to the second peak (Figure 7h). The microstructure of the precipitated phase changes when heated to the second peak at 0.9 wt.% of Li, as shown in Figure 7i, where the precipitated phase is coarser and almost completely in equilibrium η-phase. Wei et al. [35] compared the microstructure transformation diagrams of an Al-Zn-Mg-Cu alloy and a composite containing 1.0 wt.% Li using the differential thermal analysis method. Their findings also indicated that the type of aging precipitation phase of the alloy does not change after the addition of Li, which is mainly η′ phase and η phase at the grain boundaries or subgrain boundaries. However, the size of the precipitates in the Li-containing alloy is smaller than that in the Li-free alloy, and the precipitation morphology is altered, with the columnar η′ phase in the Li-free alloy and spherical η′ phase in the Li-containing alloy [69], and the addition of Li improved the stability of the η′ phase. Meanwhile, Wei analyzed the precipitation phases of the Al-Zn-Mg-Cu alloy with 1.0 wt.% Li under different aging conditions, and further concluded from the heat absorption peak of the DSC curve and the TEM morphology of the precipitation phases shown in Figure 8 that the dominant precipitation η′ phase of the alloy is not changed in different aging conditions before and after the addition of Li. They also confirmed that Li could delay the growth and coarsening speed of the precipitation phase of the GP (II) zone and η′.

When a higher amount of Li (generally more than 1.7 wt.%) is incorporated into the Al-Zn-Mg-Cu system alloy, the Li vacancy pairs formed due to the high binding energy of Li atoms and vacancies prevent the diffusion of Zn and Mg atoms to form a solute-rich GP zone, resulting in the aging process dominated by Li vacancy aggregation and almost no Zn-Mg aggregation form, thus preferentially precipitating the δ′ (Al_3_Li)-strengthening phase, which inhibits the precipitation of the original strengthening η′ phase (MgZn_2_) of the 7xxx series aluminum alloys, and instead precipitates the T ((Al,Zn)_49_Mg_32_) phase or S (A1_2_CuMg) phase or the precipitated phase X with quasi-crystalline structure [23]. The quasi-crystalline phase is a thermodynamically stable state that typically forms when excess alloying elements that cannot dissolve within the matrix are distributed at grain boundaries. A significant amount of quasi-crystalline phases are distributed in a network at the grain boundaries, which breaks the connection between the grains of the alloy, so the quasi-crystalline phase at the grain boundaries becomes the source of cracks when stressed, resulting in the rapid fracture of the alloy and a reduction in its plastic toughness. This study suggested that the incorporation of Li increases the possibility of forming quasicrystalline phases in Al-Zn-Mg-Cu alloys [75]. Wei et al. [76] studied the aging behavior of Al-Zn-Mg-Cu alloys without and with Li 1.81 wt.% and 2.12 wt.%. They found that at 100 °C aging, the lithium-free alloy exhibits the typical microstructure of Al-Zn-Mg-Cu alloys (Figure 9a), including extremely fine GP zones and η′ in the matrix. Meanwhile, coarse η and precipitation-free zones (PFZs) with a width of about 20 nm are distributed along the grain boundaries; at 160 °C aging, the coarse η′ and η phases dominate and the grain boundary precipitates coarsening, and the width of the PFZs increases to about 70 nm (Figure 9b). The Al-Zn-Mg-Cu alloys containing 1.81 wt.% and 2.12 wt.% Li display a similar precipitated microstructure, which is mainly composed of a fine δ′ phase (Figure 9c), and no precipitates are observed except for those superlattice reflections generated by δ′ particles in the electron diffraction pattern. There are a large number of δ′ phase precipitates at 160 °C peak aging (Figure 9d) and quite rough particle precipitates at dislocations and grain boundaries of the matrix (Figure 9e,f), but no uniform precipitation of the zinc-rich phase. Sodergren et al. [73] showed that in the alloys containing 2.0 wt.% and 2.5 wt.% Li, the microstructure is coarser and many of the precipitated phases are inhomogeneously nucleated at dislocations and at grain boundaries (Figure 9g,h). They concluded that these microstructures contain δ′ (AI_3_Li) phases rather than spherical η′ phase (Figure 9i), based on ordered diffraction sites with X-ray energy spectroscopy and electron diffraction analysis. Gu et al. [77] also indicated that the δ′ phase is the dominant precipitation in the aging of Al-Zn-Mg-Cu alloy containing 2.1 wt.% Li. The SAED pattern shows the superlattice reflection characteristics of δ′ at 393 K, and large quantity of fine circular δ′ particles are also identified in the dark-field TEM micrographs, with no signs of other particles throughout the aging process. In addition to the δ′ phase, the quasicrystal phase X appears in the microstructure during aging at 433 K and 473 K.

In brief, the η′ (MgZn_2_) phase is precipitated mainly at Li concentrations below 1.2 wt.%, and the δ′ (AI_3_Li) phase is formed preferentially when the Li concentration reaches 1.8–2.12 wt.%. The addition of high-content Li significantly changes the precipitation behavior of the Al-Zn-Mg-Cu alloy. The δ′ phase becomes the major precipitation phase. This is primarily ascribed to the high concentration of Li and its significant vacancy binding energy, which leads to the preferential clustering of Li atoms with quenched vacancies. As a result, the diffusion of zinc and magnesium atoms is delayed, inhibiting the homogeneous nucleation of the Zn-rich phase in the matrix, making it present mainly in the form of large particles at the grain boundaries and other crystal defects.

**Figure 9 materials-16-04750-f009:**
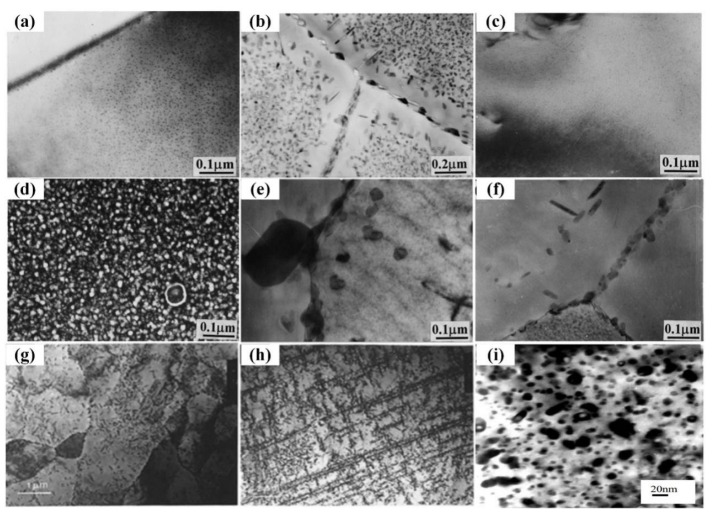
Precipitation microstructure aging at 100 °C for 76 h for: (**a**) Al-Zn-Mg-Cu alloy, (**c**) Al-Zn-Mg-Cu alloy containing 1.81% Li and at 160 °C for (**b**) Al-Zn-Mg-Cu alloy, (**d**,**e**) Al-Zn-Mg-Cu alloy containing 1.81% Li, (**f**) Al-Zn-Mg-Cu-2.12% Li alloy [76]; precipitate structure of (**g**) AA7029-2.0% Li and (**h**) AA7029-2.5% Li [73]; (**i**) precipitates in Al-Zn-Mg-Cu-1.0% Li alloys, 180 °C × 32 h [77].

In summary, due to the high vacancy binding energy of Li, the addition of Li changes the concentration of free vacancies in the Al-Zn-Mg-Cu alloy and hinders the diffusion of Zn and Mg atoms, which, in turn, changes the nucleation mode of the precipitated phase and the subsequent precipitation process. Moreover, the effect of adding different contents of Li elements to the alloy on the precipitation phase is also different; when the Li content is low, it does not change the original precipitation phase type, which is still dominated by the η′ phase. It only affects the morphology of the precipitation phase, reduces the width of the precipitation-free precipitation band, makes the dissolution activation energy of the η′ phase increase, and improves stability. However, when the Li content is high, a new metastable and ordered strengthening phase δ′ (Al_3_Li) is preferentially precipitated, which inhibits the precipitation of the original η′ strengthening phase, and instead, the coarse T((Al,Zn)_49_Mg_32_) phase and quasicrystalline phase X appear, reducing the strength of the alloy and adversely affecting the alloy properties.

#### 2.2.3. Competition Mechanism of Forming η′ and δ′ Phases

In alloy materials, precipitation competition is a frequent occurrence [78]. As pointed out above, η′ is the dominant precipitation phase during aging treatment when low Li content (generally below 1.2 wt.%) is added to Al-Zn-Mg-Cu, while in high-Li-content (generally more than 1.7 wt.%) 7xxx series aluminum alloys, η-series precipitates are completely suppressed and δ′ becomes the primary precipitation phase. Apparently, there is competition between the η′ and δ′ phases in Al-Zn-Mg-Cu-Li alloys with different Li contents.

In low-Li-content Al-Zn-Mg-Cu-Li alloys, the content of the major precipitation phase η′ is influenced by the Zn concentration in the alloy. Ji et al. [60] found that the amount of the primary age-strengthening phase η′ in 7xxx series aluminum alloys containing 1 wt.% Li is closely related to the Zn content in the alloy. As depicted in Figure 10a, the Zn/Mg ratio in the 7xxx series aluminum alloys determines the type of precipitated phase in the Al-Zn-Mg-Cu alloy and influences the forming and volume fraction of the major hardening η′ (MgZn_2_) phase. However, in 7xxx series Al alloys containing 1 wt.% Li, the Zn/Mg ratio is no more a decisive factor of the η′ equilibrium phase content. The Zn content determines the maximum amount of η′ in the alloy (Figure 10b), and raising the Zn content can boost the quantity of the η′ phase. A similar pattern was found by Zhongkui Zhao [79].

In high-Li-content Al-Zn-Mg-Cu alloys, δ′ is the dominant precipitation phase. Although the vacancy mechanism of Li could prove why δ′ precipitates faster than η-series precipitates, this does not explain the essential reason for the phenomenon that δ′ completely replaces the η-series of precipitated phases in Al-Zn-Mg-Cu alloys with a high lithium content. Wang et al. [80] showed that the interfacial energy of α-Al/η′ (Al_3_Li) (13.8 mJ m^−2^) is smaller than that of α-Al/η′ (MgZn_2_) (44.0 mJ m^−2^) [40,81]. The low interfacial energy is conducive to δ′ precipitation. According to first-principle theoretical calculations and TEM experimental observation, Liu et al. [21] revealed the initial reason why η-series precipitation is inhibited by δ′ phase and the existence of forms of Mg and Zn atoms in high-Li Al-8.58 wt.% Zn-4.38 wt.% Mg-2.88 wt.% Cu-1.26Li alloy. They showed that in spite of the lower formation of the stable precipitate η than of the δ′ (Figure 10c), the δ′ has much lower interfacial enthalpy than the η-series phase (Figure 10d). This suggests that the δ′ phase has a lower energy barrier to nucleation in the aluminum matrix and can easily achieve and maintain stability, while the η′ phase is suppressed due to the higher interface enthalpy. Moreover, in high-lithium-content Al-Zn-Mg-Cu alloy, Mg and Zn atoms readily enter δ′ precipitation and take over the positions that are normally occupied by Li and Al atoms, respectively, during thermal aging. In particular, Zn atoms have a greater tendency to ingress into the δ′ precipitates rather than remain within the matrix. With the increase in Zn atoms in the δ′ phase, the enthalpy of the zinc-doped δ′ precipitate tends to decline (Figure 10e), which provides a steady chemical structure for the δ′ phase and facilitates δ′ precipitation. Meanwhile, the formation of η-series precipitated phases is inhibited because of the very low concentration of Zn atoms in the matrix.

**Figure 10 materials-16-04750-f010:**
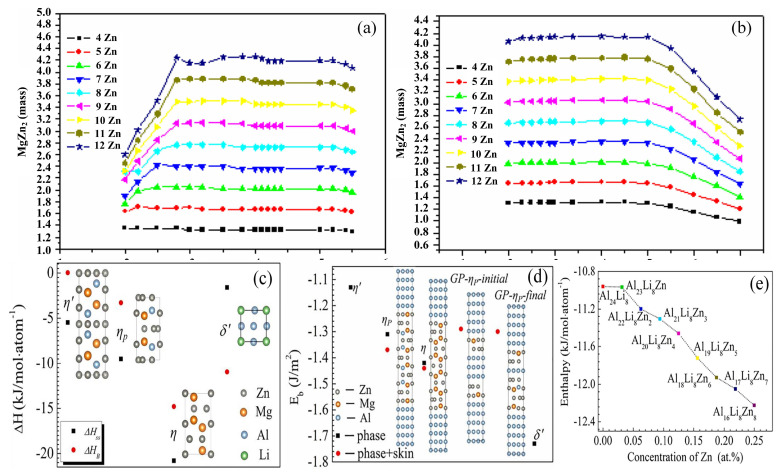
Relationship between the quantity of equilibrium-phase MgZn_2_ and the ratio of Zn/Mg in 7xxx series Al alloys (**a**) without Li and (**b**) including 1 wt.% Li [66]. (**c**) Equilibrium formation enthalpy of precipitates; (**d**) interfacial bonding energy of precipitates and “skin” containing precipitates; (**e**) plot of enthalpy of formation of Zn-doped δ′ precipitates versus Zn concentration (at.%) [21].

In summary, in the Al-Zn-Mg-Cu-Li alloy, the η′ phase precipitates preferentially as the dominant precipitating phase when the Li content is low, and its content is closely related to the Zn content within the alloy and increases as the concentration of Zn rises. When the Li content is high, the vacancy mechanism of Li and the lower interfacial enthalpy of the δ′ phase make the δ′ phase precipitate preferentially and keep absorbing Zn and Mg atoms to maintain a stable presence. Meanwhile, the precipitation of the η′ phase is suppressed by the higher interfacial formation enthalpy and the lower Zn concentration in the matrix.

## 3. Effect of Li on Properties of Al-Zn-Mg-Cu Alloys

It is well known that microalloying is an efficient strategy to improve the performance of aluminum alloys [39,82,83,84,85]. The addition of trace Li in aluminum alloy considerably reduces the alloy density and increases the elastic modulus. Meanwhile, Li, as a trace element, affects thermodynamics and aging dynamics, changes the microstructure (morphology of precipitated phase) of age-hardened aluminum alloy, and thus affects the performance of the alloy [20]. The impact of Li as a trace element on the enhanced performance of aluminum alloys has been studied. For example, Polmea [86] indicated that the incorporation of 1 wt.% Li to an Al-Cu-Mg alloy could promote the fine dispersion of the transition precipitated phase and make the alloy exhibit significant age hardening. Duan et al. [39] showed that the inclusion of 0.5 wt.% Li to AA2024 alloy can enhance the alloy hardness without decreasing the ductility of unstretched and stretched alloys (Figure 11a), and the hardening effect is more evident during aging (Figure 11b). This proves the effective effect of the microalloying of Li in 2024 aluminum alloy and also shows that the strengthening effect of Li elements on the alloy is mainly due to the modified precipitation phase instead of the solid solution reinforcement of Li. For the Al-Zn-Mg-Cu alloy, the above study shows that the addition of Li affects the age-hardening response of the alloy, and the content of Li changes the type of precipitation phase and affects the microstructure and precipitation-free precipitation zone. The precipitation strengthening is the primary reinforcement pathway for alloys; that is, the type, size, and particle spacing of the precipitation phase affect the alloy strength, and the width of PFZs is critical factor affecting the alloy ductility. Therefore, the addition of Li is bound to have a significant impact on the mechanical properties and other service properties of the Al-Zn-Mg-Cu alloy, including wear resistance and fatigue resistance, and the different Li contents lead to different major precipitation phases, which have different effects on the properties of the alloys.

**Figure 11 materials-16-04750-f011:**
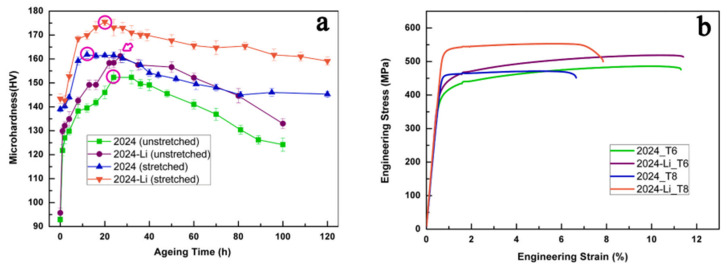
(**a**) Hardness curves versus aging time for four alloys aged at 180 °C. The circles and arrows indicate the peak aging (PA) state, respectively; (**b**) engineering stress–strain curves for the alloys in the PA state [39].

### 3.1. Mechanical Properties 

In Al-Zn-Mg-Cu-Li alloys with low Li content, the primary strengthening phase of the alloy is the finely dispersed η′ (MgZn_2_) phase. The addition of Li slows down the growth and coarsening process of the precipitated phase of the alloy [35,87]. It refines the η′ phase and improves the η′ phase stability. Additionally, Li can reduce the size and quantity of crystal boundary precipitation phases by inhibiting the diffusion of zinc and magnesium atoms, limiting solute segregation [20], and obtaining narrower precipitation-free precipitation band PFZs [16,88], thus enhancing the alloy’s mechanical performance. The enhancement of the alloy’s properties is not evident in primary aging due to the small amount of GP zone and η′ precipitates, which cannot reach the maximum hardening effect [89,90]; however, secondary or multistage aging can enhance the alloy strength hardness and improve the alloy fracture toughness without significantly reducing the ductility [39]. Huang et al. [91] found that the addition of 0.7 wt% Li to 7075 alloy resulted in better stability with increasing aging time for Li-containing alloys, especially after 360 h, when the aging hardness was higher than that of the 7075 alloy. Xie et al. [92] showed that the hardness of Al-Zn-Mg-Cu alloys containing 0.65 wt.% Li changes with approximately the same pattern as that of Li-free alloys in terms of time of aging. However, the addition of Li elements hinders the diffusion of Zn and Mg, postponing the transformation from the η′ phase to the η phase, which causes the hardness to maintain a prolonged peak time. After exceeding the peak due to the partial dissolution or transformation of the η′ (MgZn_2_) phase into η, the strengthening mechanism is changed and the hardness decreases with the development of age [88]. Wei et al. [93] studied an Al-Zn-Mg-Cu alloy with approximately 1.0 wt.% Li. As seen from the aging curves, under the same aging process, the aging response of the Li-containing alloy is significantly weaker than that of the non-lithium alloy (Figure 12a), but the hardening effect of the Li-containing alloy is obviously enhanced if it is first pre-aged at 120 °C or below and then second-grade aged at 140 °C or 160 °C. After two steps of aging, the hardening effect of the Li-containing alloy is equal to that of the Li-free alloy, as shown in Figure 12b. Wang et al. [80] showed that the microhardness and strength of the 7075–1.2 wt.% Li-Sc-Zr alloy were improved after single-stage aging and further increased upon double-stage aging, as shown in Figure 12c–f. Zhao et al. [94] studied an Al-Zn-Mg-Cu alloy including 1.1 wt.% Li. They also found that, after double-stage aging or multi-stage aging treatment, the strength of the alloy is enhanced significantly and comparable to that of 7075-T6 alloy, yet lower than that of 7075-T6 alloy. In addition, Zhou et al. [20] revealed that at peak aging circumstances, the density of the Al-Zn-Mg-Cu alloy containing Li (1.01 wt.%) decreased by 3.7% after secondary aging, and the elongation was nearly 9%. The alloy exhibits more than 20% higher ultimate tensile strength than that of the Al-Zn-Mg-Cu alloy without Li, and 28% higher specific strength than that of the 7075 alloy, with no significant decrease in ductility. Moreover, the addition of 1.01 wt.% Li promotes the precipitation of zinc-rich and magnesium-rich phases within the matrix, diminishes the formation of large zinc-rich particles at grain boundaries, and decreases the intergranular fracture susceptibility of the lithium-containing alloy. The fracture surface is characterized by a mixture of through- and along-crystal, and the fatigue crack extension resistance is also improved at moderate stress intensity levels. Second-stage or multistage aging has a remarkable effect on improving the properties of the alloy. Wei et al. [93] attributed this phenomenon to the fact that after one step of the aging of Li-containing alloys, the GP zone becomes smaller and cannot reach the maximum hardening effect, while in two-step aging, both GP zones formed in pre-aging are maintained at the temperature at which the second aging step is performed, and enough η′ phase precipitates are formed to give a higher aging hardness. Bai et al. [89] showed that the main precipitated phase of 7075 + 1 wt.% Li at one-step aging is the GP zone, while at two-step aging, the peak aging appears as the η′ phase, which increases the hardness of the alloy. Zhao et al. [94] also came to a similar conclusion; they pointed out that the main precipitates of Al-Zn-Mg-Cu alloy including 1 wt.% Li at one-step aging are the GP(I) zone with a small amount, while at two-step aging, the main precipitates are a mixture of GP zone and η′ phase with a larger amount, which has a good strengthening effect on the alloy. Meanwhile, they also showed that, in the 7xxx series aluminum alloy containing Li, even though the quantity of η′ precipitates is comparatively small, the performance can be improved to a level comparable to that of the 7xxx series alloys by multi-stage aging at 120 °C peak aging. The multistage aging strengthening mechanism is shown in Figure 13. Therefore, Al-Zn-Mg-Cu-Li alloys with low lithium content can achieve excellent specific strength, modulus, and plasticity at low density after suitable heat treatment.

At higher Li content in Al-Zn-Mg-Cu-Li alloys, δ′ is the main precipitating phase, and with the gradual increase in the Li content, an increase in the δ′ concentration and a decline in the strength, toughness, and ductility of the alloy are noted [20]. Several researchers have pointed out that the unfavorable embrittlement of alloys containing >1.7 wt.% Li is noted due to Li segregation at grain boundaries, causing new δ′ phase precipitation and changes in the width of the δ′ precipitation-free zone (PFZ). The δ′ phase has a slight mismatch with the Al matrix and is easily penetrated by mobile dislocations, resulting in a relatively low yield strength. The formation of a high-volume fraction of δ′ leads to planar slip in Al-Zn-Mg-Cu alloys with a high Li content under plastic deformation, generating significant local stress concentrations, thus reducing the ductility [21,95,96]. Moreover, with the increase in the δ′ concentration, the content of the GP (Zn, Mg) zone decreases, and the Zn-rich and Mg-rich phases precipitate mainly in the form of large particles at the grain boundaries, which also reduces the elongation, changes the fracture mode, and increases the intergranular fracture tendency of the alloy. Huang et al. [27] and Sodergren et al. [73] separately investigated the impact of 2.0–2.5 wt.% Li incorporation on the aging responses of 7075 and 7029 alloys; they showed that the δ′(Al_3_Li) phase is the only strengthening phase in the Al-Zn-Mg-Cu-Li alloy and the coarse T [(Al,Zn)_49_ Mg_32_] phase replaces the fine η′ inhomogeneous precipitation with almost no strengthening effect. The addition of Li did not increase the strength of the matrix alloy, but Li significantly affected the tensile elongation of the alloy. Wei et al. [76] found that Al-Zn-Mg-Cu alloys containing 1.81 wt.% and 2.12 wt.% Li had significantly lower age-hardening rates than the Li-free alloys under single-stage aging at 100 and 120 °C due to the nucleation and slow growth of the δ′ phase. The single-stage aging of Li-containing alloys is mainly reinforced by the δ′ phase, which exhibits a slight mismatch with the aluminum matrix and is susceptible to being severed by dislocation motion, resulting in a relatively lower yield strength. Simultaneously, the incorporation of Li resulted in the precipitation of the Zn-rich phase predominantly as large particles along the grain boundary in the alloy, which altered the fracture behavior of the alloy from transgranular fracture (Figure 12g) to intergranular fracture (Figure 12h), leading to an enhanced propensity for intergranular fracture and a decrease in elongation, and the two-step aging process did not significantly improve strength or elongation. However, they also found that pre-deformation before aging can promote the heterogeneous nucleation of Zn-rich phases in the matrix, increase the concentration of defects in the matrix, and change the size and distribution of Zn-rich precipitates. Al-Zn-Mg-Cu alloys with 1.81 wt.% and 2.12 wt.% Li after 20% pre-deformation exhibit comparable tensile properties to 7075 alloy (T6) and possess a 7% lower density, which holds great promise for various applications in the aerospace field.

**Figure 13 materials-16-04750-f013:**
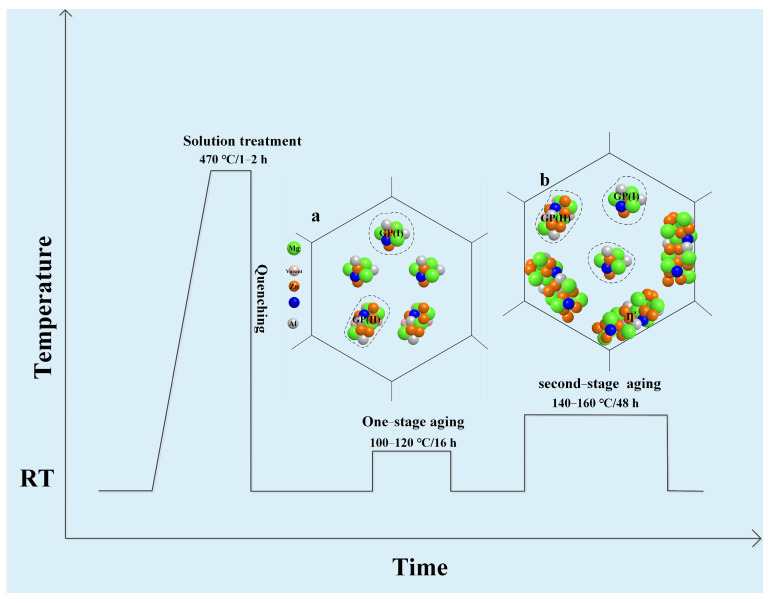
One−stage and second-stage aging strengthening mechanism of Al−Zn−Mg−Cu alloys containing lithium: (a) precipitation process during one-step aging and (b) precipitation process during two−step aging [45,66,97,98].

In summary, when the Li content in Al-Zn-Mg-Cu-Li alloy is high and δ′ becomes the main precipitation phase in the alloy, the uniform precipitation of the η′ phase is inhibited during aging, leading to unsatisfactory mechanical properties. Although pre-deformation can considerably enhance the yield strength of the alloy while maintaining good elongation, the optimal amalgamation of the precise amount of deformation and the temperature of aging remains uncertain. However, at low Li content in Al-Zn-Mg-Cu-Li alloys, the finely dispersed η′ (MgZn_2_) phase can improve the properties of the alloy, and especially at secondary or multi-stage aging, Li can increase the alloy’s strength and hardness and improve the fracture properties without significantly reducing the ductility. Nevertheless, it should also be noted that, if the Li content is too low, the η′ size becomes larger, which also diminishes the benefits of the low density and high specific strength of lithium-containing alloys. Therefore, the Li content and heat treatment process of 7xxx series alloys must be under stringent control to obtain a uniformly dispersed fine η′ phase rather than coarse δ′ phase in order to obtain good overall performance of the Al-Zn-Mg-Cu-Li alloy.

### 3.2. Friction and Wear Behavior

Wear is a significant form of damage that arises on the surface due to friction between moving parts in contact. Furthermore, it is a prevalent issue observed across various industrial applications, leading to a decrease in system efficiency [74,99,100,101,102,103]. It is widely recognized that there exists a definite correlation between the hardness and wear resistance of the materials [104,105]. The addition of Li, as noted above, affects the hardness of the alloy and therefore necessarily affects its frictional wear behavior. Xie et al. [106] prepared 7075 (0 wt.%, 0.5 wt.%, 1 wt.%, 2 wt.%)-Li alloy using the hot pressing sintering method by mixing the raw materials on a TD-6 three-dimensional mixer and sintering the ball-milled powder in a graphite mold on a medium-frequency self-controlled hot press. The sintered samples were tested for room temperature friction and wear properties using an MFT-5000 tribometer to study the impact of Li on the microstructure, friction, and wear behavior of the 7075 aluminum alloy. Their results indicated that the friction factor of 7075-Li alloy generally shows a trend of first increasing and then decreasing to a steady trend. The friction factors of 7075-Li alloys are 0.548, 0.524, 0.580, and 0.578, respectively. When the Li content is ≥1 wt.%, the friction factors of the lithium-containing Al alloys are higher than those of the Li-free Al alloys, but the friction factors of 7075-2 wt.% Li alloys are smaller than those of 7075-1 wt.% Li alloys (Figure 14). The wear of the Li-containing alloy increases as the Li content increases. The researchers used FE-SEM to characterize the wear sample surface and wear chips and used an MFP-D white light interferometer to observe the surface morphology of the specimen and found that the wear scar cross-section of 7075 aluminum alloy is similar to a trapezoid, with a wide and deep wear scar and large bottom. The 7075-0.5 wt.% Li alloy has a similar width to 7075 aluminum alloy, but with a larger depth. When the Li content is ≥1 wt.%, the width and depth values of the Li-containing alloy are larger than the corresponding values of Li-free aluminum alloy (Figure 14D). The wear scar cross-sections of 7075-0.5 wt.% Li and 7075-1 wt.% Li alloys are similar to a triangle: the wear scars are wide and deep with a narrow bottom, while the wear scar cross-section of 7075-2 wt.% Li alloy is similar to a trapezoid, with the bottom of the wear scar being flat and wider than other alloys, and the wear resistance is poor.

Moreover, as seen from the SEM image of the alloy wear marks (Figure 14B), the surface of the 7075 aluminum alloy shows flaking areas, plastic deformation areas, wear chips, and small furrow bands. The 7075-0.5 wt.% Li alloy shows a significant decrease in flaking areas, an increase in furrows, and a widening of the alloy, as well as a small number of particles and delamination, indicating abrasive wear and adhesive wear. The 7075-1 wt.% Li alloy shows massive flaking areas, a small amount of furrows, and a significant increase in delamination, which are characteristic of adhesive wear, indicating a decrease in frictional properties and an increase in wear. When the Li content was further increased to 2 wt.%, spalling zones appeared on the surface of the 7075-2 wt.% Li alloy and adhesive wear was accompanied by plastic deformation. The 7075-2 wt.% Li alloy is prone to flaking areas, delamination, and craters under load due to its low hardness and cold-welding action, and the flaking chips are coated on the grinding balls to act as solid lubrication [107]. This also explains the lower friction factor of the 7075-2 wt.% Li alloy compared to the 7075-1 wt.% Li alloy. Meanwhile, as the friction time increases, the temperature of the contact surface rises and oxidation occurs in the alloy, and oxidative wear exists to a degree in all 7075-Li alloys [108].

In addition, as seen from the SEM pattern of the 7075-Li alloy chips (Figure 14C), the appearance of the chips is flaky, and with the increase in Li content, a mixture of granular and flaky chips are formed in the Al-Li alloy. Finally, the chips are mainly thick lamellae, and their width and length increase significantly compared with those of Li-free aluminum alloy. When Li is not added, the hardness of the aluminum alloy is low, and the surface is cold-welded with a ceramic ball. As such, the material flakes off from the surface and adhesion wear occurs. When the Li content is 0.5 wt.%, the aluminum–lithium alloy is harder, the material is less likely to flake under load, and the alloy’s wear resistance has improved. With the further increase in the Li content of the Al-Li alloy, oxidation becomes severe, brittle, and hard and an Al_2_O_3_ wear layer is formed on its surface, resulting in greater shear strain on the alloy, while the Al_2_O_3_ wear layer is brittle and weakly bonded with the Al-Li alloy matrix, making it easy to peel off under the action of load, revealing new material on the matrix. Moreover, as the Li content increases, the η′ MgZn_2_ phase decreases, its protective effect on the Al-Li alloy decreases, the hardness of the Al-Li alloy decreases, the intermetallic phase in the microstructure of the Al-Li alloy increases, the dendritic spacing increases, and the material is more easily flaked [109]. Therefore, when the Li content is ≥1%, in addition to the occurrence of oxidation wear, the Al-Li alloy is prone to adhesive wear and reduced wear resistance is noted. Overall, the 7075-0.5 Li alloy has high hardness, a small friction factor, a low wear rate, and better wear resistance.

### 3.3. Fatigue Resistance

Fatigue refers to the fracture failure of structural parts under the action of long-term cyclic stress or strain, which is one of the critical design concerns in structural components and is widely recognized as the primary cause of failures [110]. The Al-Zn-Mg-Cu alloys are exposed to metal fatigue due to cyclic loading, corrosion under environmental conditions, and thermal shock from abrupt temperature fluctuations in transmission systems. The formation and development of fatigue cracks caused by cyclic loading is one of the most significant problems in these alloys [111,112]. From the data collected in the literature, e.g., Refs. [113,114,115], the Al-Li alloy has significantly higher fatigue strength than the common Al alloys 2024 and 7075. The beneficial effects of Li addition on high-cycle fatigue (HCF) resistance (i.e., HCF strength) of high-strength aluminum alloys have well been recognized (Figure 15a) [70]. Di et al. [116] investigated the influence of Li addition (1.82 wt.%) on the HCF resistance of aluminum; they found that Li addition greatly enhanced the fatigue strength of both the solution-treated and aged specimens. Yang et al. [117] investigated the influence of trace additions of Li and Zn on the fatigue crack growth rate of Al-4.2Cu-1.4Mg alloy. They used the Kahn tearing test method for fracture toughness, testing on an Instron electronic tensile machine and the Schenck-6T hydraulic servo-programmed fatigue test machine for fatigue crack expansion rate testing in accordance with GB/T 6398-2000 “Fatigue crack expansion rate test method for metallic materials” (sine wave, frequency of 10 Hz, room temperature, atmospheric environment). The results revealed that the addition of Li and Zn elements significantly improves the fracture toughness and reduces the fatigue crack expansion rate of the alloy, and the fracture toughness is improved more and the fatigue crack expansion rate of the alloy is slower with the increase in the Li addition. Furthermore, the fracture toughness and fatigue crack expansion rate of the alloys with Li and Zn added together (A2, A3) demonstrate more significant change than those of alloys with Li added alone (A4), as shown in Figure 15b. For the Al-Cu-Mg alloy, the addition of trace amounts of Li and Zn elements reduces the grain size and allows for a more uniform distribution of the excess and diffuse phases in the alloy after heat treatment, which is the microscopic mechanism for the improved fatigue resistance of this alloy. In the case of the Al-Li alloy, since its main precipitated δ′ phase is diffusely distributed in the alloy, its easy shear property may lead to reversible slip, which reduces the generation of dislocations and other defects (dislocations, slip, and other defects in the areas where fatigue cracks are easy to sprout and expand) and alleviates the internal local stress–strain concentration under the action of external forces. As a result, Al-Li alloy has a longer fatigue crack initiation life and, thus, a higher fatigue life owing to its internal structure. In addition, some other studies on the fatigue resistance of alloys [118,119,120,121,122,123,124] showed that the precipitation phase, grain size, aging degree, and heat treatment process all affect the fatigue resistance of the alloy to some extent. According to the above study, the addition of Li affects the grain size and microstructure of the alloy and the Li content affects the type of precipitated phase of Al-Zn-Mg-Cu alloy. It can be seen that the effect of Li addition to Al-Zn-Mg-Cu alloy on the fatigue resistance of the alloy is the result of a series of factors. However, since the effect of Li content on fatigue strength cannot be accurately distinguished [125], there is no research on the quantitative relationship between Li content and fatigue life.

## 4. Summary and Outlook

This article reveals the role of Li in the aging precipitation process of Al-Zn-Mg-Cu alloy. Because the element Li has a higher vacancy-binding energy than Zn or Mg, it preferentially captures vacancies to form Li vacancy clusters, reduces the concentration of free vacancies, and promotes the aggregation of Zn and Mg atoms, thus promoting the formation of the GP(I) zone and inhibiting the formation of the GP(II) zone and the coarsening of the η′ phase.

The effect of Li content on the precipitated phases in Al-Zn-Mg-Cu alloys and the mechanism are explained. With different Li content, the precipitation competition between the η′ and δ′ phases occurs. When the Li content is low, the primary precipitation phase in Al-Zn-Mg-Cu-Li alloys remains the η′ phase and its content increases with the increase in Zn content. While the precipitation phase type changes at a higher Li content, the vacancy mechanism of Li and the lower interfacial energy of the δ′ phase promote the preferential precipitation and stable existence of the δ′ phase, so that the precipitation of the original strengthening phase η′ is suppressed, and coarse T ((Al,Zn)_49_Mg_32_) and quasicrystalline phases also appear, which adversely affect the alloy’s properties.

In addition, the impact of Li addition on the mechanical performance and fatigue and wear resistance of the alloy are also elaborated. In the case of low Li content, η′ is the dominant precipitation phase. Li only affects the morphology of the precipitation phase, reduces the width of the precipitation-free zone, and enhances the stability of the η′ phase, thus improving the mechanical properties of the alloy. In particular, secondary or multistage aging can enhance the specific and ultimate tensile strength of the alloy without significantly reducing its ductility, improve the fracture mode, and reduce the tendency of intergranular fracture [39]. In contrast, the precipitation of the δ′ phase at a high Li content reduces the strength and elongation of the alloy and deteriorates its mechanical properties. Moreover, the alloy with low Li content (about 0.5 wt.%) has a low friction factor, small wear, triangular wear scar cross-section, wide and deep wear scar, and narrow bottom, showing the characteristics of abrasive wear and adhesive wear, with good wear resistance. However, when the content of Li is ≥1 wt.%, the friction factor of the aluminum alloy containing Li increases, the Li-containing aluminum alloy is prone to adhesive wear, and the wear resistance decreases. 7075-0.5 Li alloy has better wear resistance. In addition, the addition of moderate amounts of Li elements can also improve the fatigue strength and fracture toughness of the alloy, reduce the fatigue crack expansion rate, slow down the fatigue crack expansion of the alloy, and improve the fatigue life of the aluminum alloy to a certain extent. Trace amounts of Li improve the fatigue resistance of this alloy by reducing the grain size and making the distribution of diffusion more uniform, while high levels of Li influence the fatigue resistance of the material by affecting the type of precipitated phase in the Al-Zn-Mg-Cu alloy. However, fatigue strength is the result of the combined effect of multiple factors, and the effect of Li content has not been accurately distinguished yet.

7xxx series aluminum alloys are extensively utilized in aerospace, and further lightweighting, increasing the modulus of elasticity, improving the fatigue resistance and corrosion resistance of the alloy, and thus extending the service life of the material are the main development directions at present [126]. Li is very effective for developing lightweight aluminum alloys and improving their elastic modulus, and it is a new and effective path to further enhance the performance of aluminum alloys by using the micro-alloying effect of Li. For high-strength aluminum alloys strengthened by heat treatment, such as Al-Zn-Mg-Cu, it is of great significance to control the Li content and heat treatment process and then improve the precipitation enhancement effect of the alloy, in order to improve the performance of alloys. However, there is currently a dearth of systematic studies on the impact of Li content on the aging precipitation process and precipitation phase of the alloy, and the critical value of Li content in the δ′ phase replacing the η′ phase in the Al-Zn-Mg-Cu-Li alloy and the optimal values of Li content are not clear and need further in-depth study. Moreover, with the development of aluminum–lithium alloys in recent years, many researchers have proposed that a fourth-generation aluminum–lithium alloy with excellent performance to be developed in the next step also has the tendency to further reduce the Li content [127], which shows that an Li-containing aluminum alloy with a low Li content has broad development prospects and research value. The addition of a low amount of Li in 7xxx series alloys not only improves the mechanical properties such as hardness, specific strength, and ultimate tensile strength and changes the friction factor and wear characteristics, thus improving wear resistance, but also plays a positive role in improving the fatigue strength and fracture toughness and extending fatigue life. Therefore, one future research direction will be to study the role of Li in aluminum alloys and the influence of Li content on the service performance of high-strength Al-Zn-Mg-Cu alloys, especially the effect of Li on fatigue resistance and corrosion resistance [128], so as to find lightweight and high-strength aluminum alloys containing Li with excellent properties. In addition, the preparation process of lithium-containing aluminum alloys is also an urgent problem and research hotspot in this field because the chemical nature of lithium is exceptionally active and it is extremely easy to oxidize and burn, so it is difficult to add to aluminum alloy using conventional methods. 

## Figures and Tables

**Figure 1 materials-16-04750-f001:**
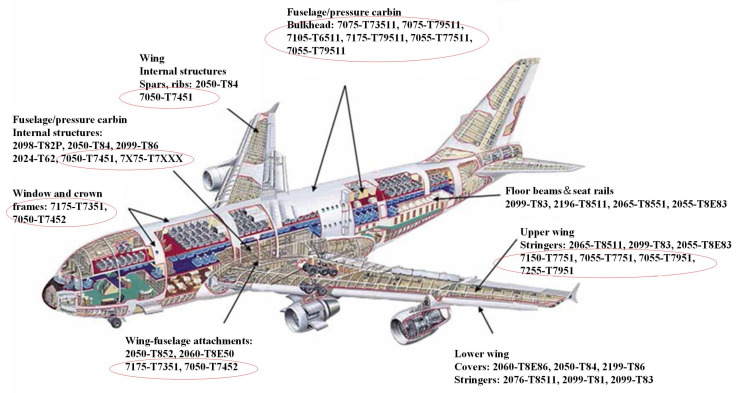
Examples of applications of 7xxx series aluminum alloys used in the aircraft field [1].

**Figure 7 materials-16-04750-f007:**
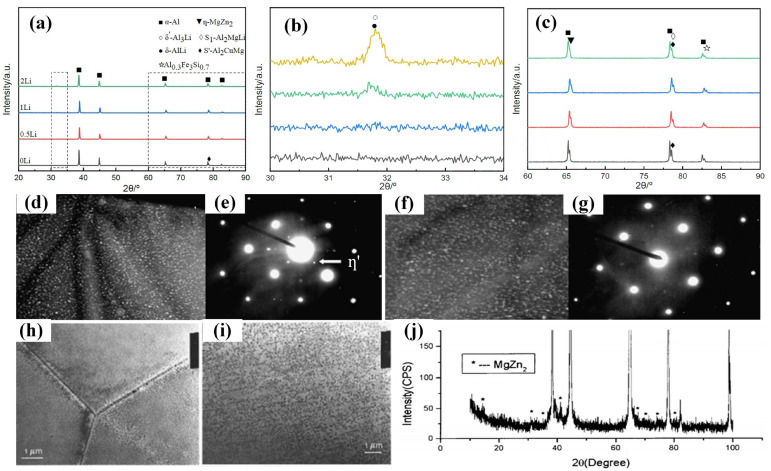
(**a**) XRD patterns and (**b**,**c**) local enlarged patterns of 7075 aluminum alloy with different Li contents [70]; precipitates images and corresponding SAED patterns of 7055-1 wt.% Li alloys: (**d**,**e**) after aging 24 h at 120 °C, (**f**) and (**g**) after aging 28 h at 160 °C [72]; structures of the precipitated phase of (**h**) AA7029-0.4% Li and (**i**) AA7029-0.9% Li [73]. (**j**) XRD pattern of Al-Zn-Mg-Cu-1.01 wt.% Li alloy [20].

**Figure 8 materials-16-04750-f008:**
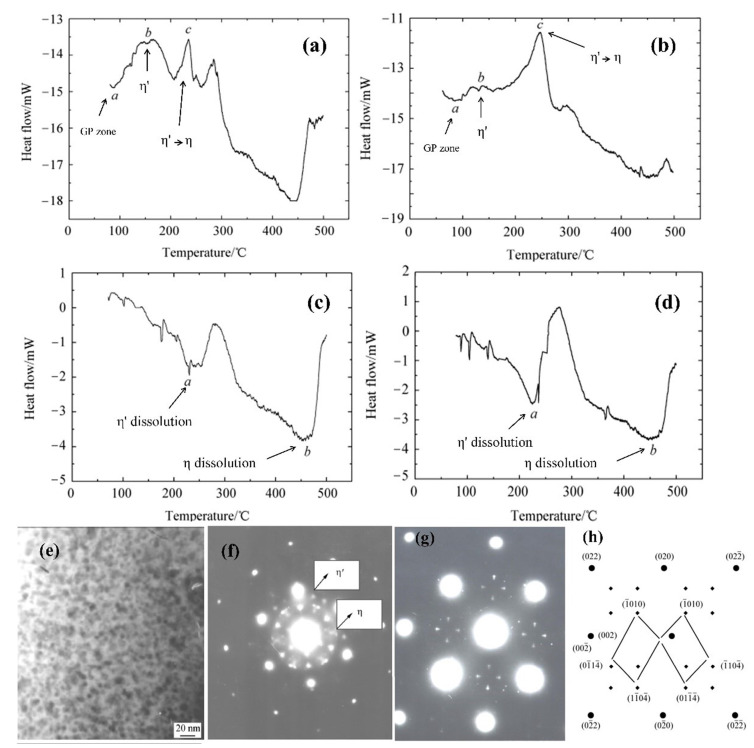
DSC curves of (**a**) Al-Zn-Mg-Cu alloy and (**b**) the alloy containing 1 wt.% Li under solution treatment; DSC curves of Al-Zn-Mg-Cu alloy containing 1 wt.% Li aged at (**c**) 160 °C for 48 h and (**d**) 180 °C for 32 h; (**e**) TEM microstructure and (**f**) SAED patterns of precipitated phase of Al-Zn-Mg-Cu alloy containing 1 wt.% Li after 48 h aging at 160 °C; (**g**) SAED and (**h**) [74] Al plane diffraction patterns of 1 wt.% Li Al-Zn-Mg-Cu alloy aged at 180 °C for 32 h [35].

**Figure 12 materials-16-04750-f012:**
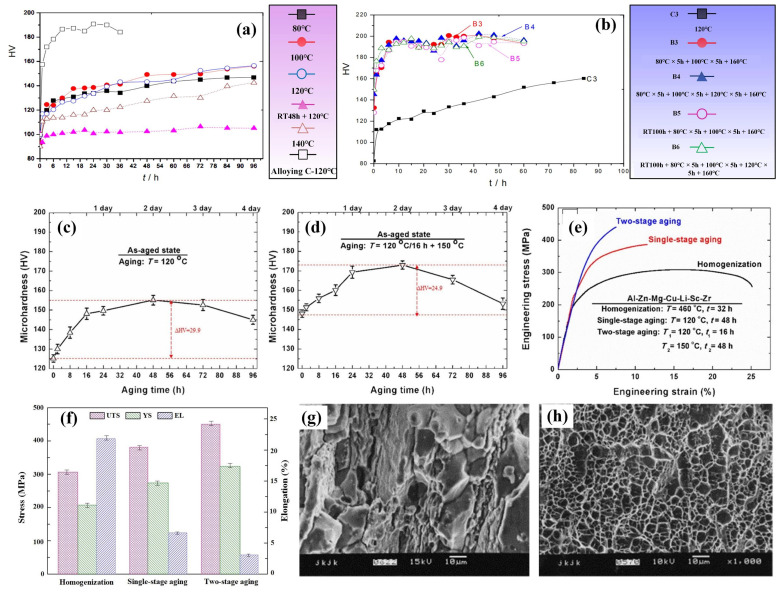
(**a**) Hardness curve of Al-Zn-Mg-Cu alloy without Li at 120 °C, 120 °C, 120 °C, 140 °C, or 48 h with 1.1 wt.%Li; (**b**) effect of two-step aging on the hardening effect of Al-Zn-Mg-Cu alloy containing 1.1 wt.% Li [93]. Microhardness changes of Al-Zn-Mg-Cu-Li-Sc-Zr (A_1_) alloy (**c**) aged at 120 °C and (**d**) aged at 150°C previously aged at 120 °C/16 h. (**e**) Engineering stress–strain curves of A_1_ alloy; (**f**) tensile test results of A_1_ alloy [80]. (**g**) Fracture morphology of peak-aging Al-Zn-Mg-Cu alloy; (**h**) fracture morphology of peak-aging alloy Al-1.81 wt.% Li-Zn-Mg-Cu alloy [76].

**Figure 14 materials-16-04750-f014:**
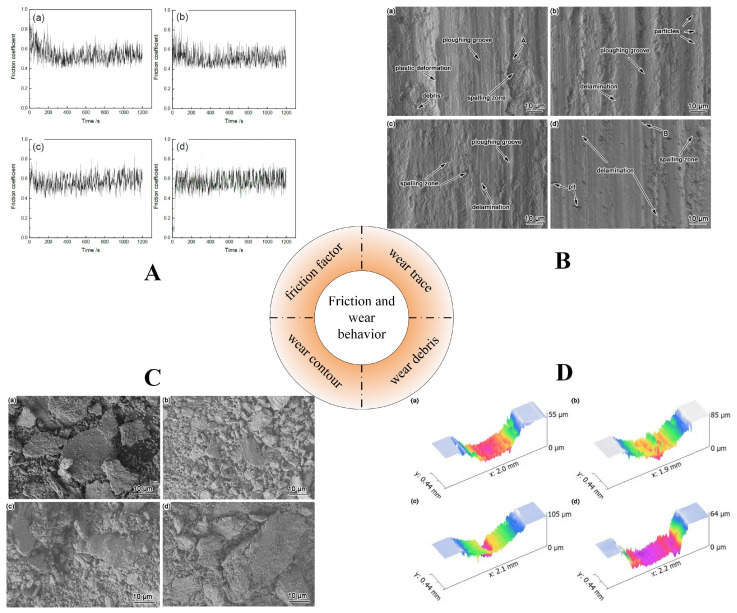
Frictional properties of 7075 Al alloy with varying lithium contents ((**a**–**d**) represent 0 wt.%, 0.5 wt.%, 1 wt.%, and 2 wt.% of Li, respectively): (**A**) friction coefficient; (**B**) SEM of wear marks; (**C**) SEM of wear debris; (**D**) 3D view of friction surface [106].

**Figure 15 materials-16-04750-f015:**
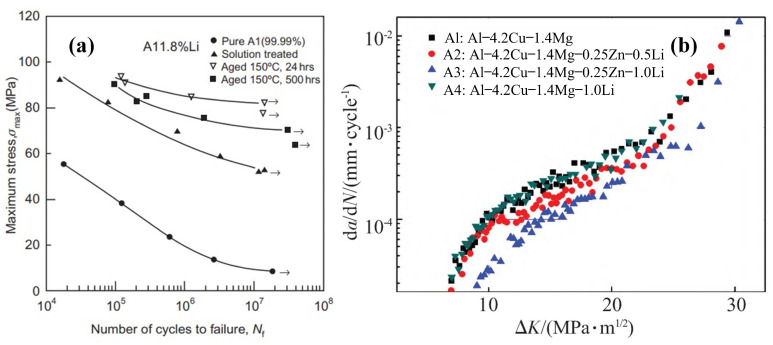
(**a**) HCF smooth specimen R521 data for an Al 1.82 wt.% Li binary alloy compared to pure aluminum [70]. (**b**) Comparison of fatigue crack expansion rates of Li-containing alloys [117].

## Data Availability

Not applicable.

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
