# Peer review of "The Role of Lithium in the Aging Precipitation Process of Al-Zn-Mg-Cu Alloys and Its Effect on the Properties"

_materials, 2023, doi:10.3390/ma16134750_

Round 1
Reviewer 1 Report
R E V I E W
ID-2464240 /09.06.2023
“The Role of Lithium in the Aging Precipitation Process of Al-Zn-Mg-Cu Alloys and its Effect on the Properties”
Detailed observations
Line 85-86- The purpose of the review is not clear. What is offered to the reader? A short review of articles in the field or another purpose is pursued. It must be specified, especially that gaps were observed in the conclusions.
Fig.1 – How is it correct: "Example applications" or "Examples of applications"?
Line 95 – "solid solution" is not a treatment but a physical state. “Putting into solution” is the treatment.
Fig.4 – Put small letters (a, b, c...) as in the figure and in its explanatory text (there you put A, B, C...).
Line 187-188 - Use "transition" twice and the phrase becomes unclear. Rephrase.
Line 192-193 -the word “metastable” is separated incorrectly, and “transpire” must be replaced with a technical word.
Line 202 – It is said that the phase Æž has a hexagonal structure but in Fig. 5-b the phase Æž' is presented as hexagonal. Correct or provide additional explanations.
Line 213-214 – The phrase is not clear. What is the connection between Al3Li and (Cu3Au)???
Line 231 - Is it δ or δ'???
Line 238 – In Fig. 4E there are not 162 atoms to be invoked.
Line 260 – Correct Fig. 4B with Fig. 4b.
Line 264-267 – So far we have talked about the δ phase. Suddenly δ' phase appears without any explanation.
Line 284 – The θ phase (Al2Cu) was not discussed in 2.2.1. It should at least be mentioned.
Line 293-294 – You have twice δ'!!!
Fig. 10 - Diagram (c) does not have the power (-1) on the ordinate as diagram (e) does. Where is it wrong?
Fig. 13 – It reminds us that the work lacks the aging diagrams for each commented phase. And in this figure13 neither temperature nor time values appear on the axes
For figures - Some have not mentioned the source, e.g. Fig. 3, 4, 13. Tab 1 does not have the source indicated either.
Summary and Outlook – All essential observations from the text are incompletely repeated. In a REVIEW, the systematized opinions of the authors must appear. In addition, the current form of the text does not even cover all the main aspects of the paper (e.g. the evolution of mechanical, corrosion, and wear properties). The material must offer to the readers a systematized opinion, useful in practice or in research.
General Observation – Both the alloying technology with Li, as well as the machines and equipment used by the authors, mentioned in the paper, must be briefly described so that the readers can get an idea. How was corrosion measured, how was fatigue measured, and how was wear measured?
Recommendation- The work should be revised majorly. Repetitions must be avoided (about Li limit percentages were written in the paper more than 4 times), and the information about the equipment, about the aging diagrams, must be completed, because the title of the paper requires this, and with the authors' systematized conclusions, useful to the reader.
Reviewer 2 Report
Thanks to the authors for the work done in creating such a review of the literature. To be honest, I do not know how to evaluate such works, in my humble opinion, there is no particular need for works of this format. Authors involved in the development of new groups of alloys must learn and understand such things at the very beginning of their scientific path. However, if the authors do similar work, then it can be useful to someone.
To be frank with the editor and authors, I agreed to the review of this work by mistake, since there is no clear mention in the title and abstract that this work is a review.
1) Please correct the title of this work so that it would clearly indicate that this work belongs to the review.
2) it was somewhat surprising that the literature references contain such a small number of modern literature sources (I may be wrong, but only 10 references refer to the year 2020+)
3) perhaps this is just an observation and I am seriously mistaken, but the authors focus on the work of scientists from China. If this is not the case, then I apologize.
4) Please check the spelling of compound formulas in the entire work additionally. L12 It is customary to write down the number "2" with a subscript. Also for P63/mmc and others.
5) I didn't understand how you used link 103 to your discussions.
Round 2
Reviewer 1 Report
All of the modifications mqde cover the observations.
Congratulation!